# Potential of Sulforaphane and Broccoli Membrane Vesicles as Regulators of M1/M2 Human Macrophage Activity

**DOI:** 10.3390/ijms231911141

**Published:** 2022-09-22

**Authors:** Tamara Ramírez-Pavez, Andrea García-Peñaranda, Paula Garcia-Ibañez, Lucía Yepes-Molina, Micaela Carvajal, Antonio J. Ruiz-Alcaraz, Diego A. Moreno, Pilar García-Peñarrubia, María Martínez-Esparza

**Affiliations:** 1Biochemistry and Molecular Biology (B) and Immunology Department, School of Medicine, IMIB and Regional Campus of International Excellence “Campus Mare Nostrum”, Universidad de Murcia, 30100 Murcia, Spain; 2Aquaporins Group, Centro de Edafología y Biología Aplicada del Segura (CEBAS-CSIC), Campus de Espinardo, 30100 Murcia, Spain; 3Phytochemistry and Healthy Food Lab (LabFAS), Department of Food Science Technology, Centro de Edafología y Biología Aplicada del Segura (CEBAS-CSIC), Campus de Espinardo, 30100 Murcia, Spain

**Keywords:** inflammation, phagocytosis, macrophage, plant membrane vesicles, sulforaphane, hormesis

## Abstract

Macrophages have emerged as important therapeutic targets in many human diseases. The aim of this study was to analyze the effect of broccoli membrane vesicles and sulphoraphane (SFN), either free or encapsulated, on the activity of human monocyte-derived M1 and M2 macrophage primary culture. Our results show that exposure for 24 h to SFN 25 µM, free and encapsulated, induced a potent reduction on the activity of human M1 and M2 macrophages, downregulating proinflammatory and anti-inflammatory cytokines and phagocytic capability on *C. albicans.* The broccoli membrane vesicles do not represent inert nanocarriers, as they have low amounts of bioactive compounds, being able to modulate the cytokine production, depending on the inflammatory state of the cells. They could induce opposite effects to that of higher doses of SFN, reflecting its hormetic effect. These data reinforce the potential use of broccoli compounds as therapeutic agents not only for inflammatory diseases, but they also open new clinical possibilities for applications in other diseases related to immunodeficiency, autoimmunity, or in cancer therapy. Considering the variability of their biological effects in different scenarios, a proper therapeutic strategy with *Brassica* bioactive compounds should be designed for each pathology.

## 1. Introduction

Macrophages are versatile cells belonging to the innate immune system. They include a heterogeneous cell population that induces the adaptive immunity and plays a crucial role in inflammation and tissue remodeling, among others. These cells reside in all tissues where they display a great anatomical and functional diversity [1]. In homeostasis, the tissue-specific macrophage population is composed by a mix of embryonic derived macrophages, which are maintained by self-renewal; and macrophages derived from monocytes migrated into the noninflamed tissues [1]. Circulating peripheral blood monocytes are also rapidly recruited and differentiated into macrophages in response to a great variety of local danger signals [1,2]. This process can also play a major role in the initiation and maintenance of chronic inflammatory diseases [3].

Two main phenotypes to which macrophages can polarize have been broadly described, namely the classical cell population or M1 (also known as inflammatory, microbicidal, or classical activated macrophages) and the alternative phenotype or M2 (also known as immunosuppressive, anti-inflammatory, tissue repair, or alternatively activated macrophages) [4,5]. M1 macrophages are mainly involved in proinflammatory responses; therefore, among their main functions are the production of proinflammatory mediators, such as interleukin (IL)-6, IL-1β, and tumor necrosis factor (TNF)-α, and the development of effective mechanisms for pathogen killing through phagocytosis, production of nitrogen and oxygen-reactive molecules, and antigen-presenting activity. On the other hand, M2 macrophages are mainly involved in anti-inflammatory responses, producing cytokines, such as IL-10, and repair of damaged tissues through phagocytosis of debris, damaged, dead, and apoptotic cells, which inhibits the production of proinflammatory cytokines [2]. While tissue-resident macrophages display a mixed and dynamic pattern of polarization and activation [5], under in vitro conditions, macrophages can be specifically differentiated from monocytes to a unique classical (M1) or alternative (M2) pattern by stimulation with different molecules [5,6].

Physiologically, the balance between M1 and M2 polarization and the inflammatory response must be tightly controlled, since it could cause tissue damage and contribute to its progression, as it has been described for many autoimmune [7] and chronic inflammatory diseases [1,2], including endometriosis [8,9] or hepatic cirrhosis [10]. Therefore, macrophages and inflammation have emerged as important therapeutic targets in many human diseases [1,2,5,7,8,9,10,11].

In this regard, the study of phytochemicals to manage chronic inflammation has increased in recent years. The consumption of cruciferous vegetables has shown health benefits [12,13,14] associated with the intake of glucosinolates and their derivatives, isothiocyanates, which are produced by the enzymatic action of myrosinase [15]. Sulforaphane (SFN) (R-1-isothiocyanato-4-methylsulfinyl butane) is a hydrolysis compound of the glucosinolate glucoraphanin, which is predominantly found in *Brassica oleracea* L. var. *Italica* (broccoli) [12].

SFN has been noted for its anti-inflammatory, antioxidant, apoptosis-inducing, and anticarcinogenic effects [12,13,14,15,16,17,18]. SFN is an inhibitor of the nuclear factor kappa B (NFκB) pathway, which is crucial in the transcription of proinflammatory cytokine genes in response to different stimuli [19]. It is also an inducer of the nuclear factor erythroid 2-related factor 2 (Nrf2), which is responsible for the transcription of genes involved in antioxidant activities or anti-inflammatory pathways [20,21].

SFN could be encapsulated into different types of nanocarriers to improve its stability and half-life in vitro or in vivo [22,23,24]. Membrane vesicles isolated from *Brassica* spp., such as broccoli or cauliflower, are a suitable source of membrane vesicles as nanocarriers for use in biotechnological applications [25,26,27,28]. Its content in aquaporins, which are intrinsic membrane proteins that can bind and stabilize SFN, contributes to its thermodynamic stability [29]. Therefore, broccoli membrane vesicles have been used to encapsulate SFN, improving its antiproliferative and anti-inflammatory activity in macrophage model showing two phases of release, one fast occurring through membrane lipids and the other occurring slowly after liberalization with aquaporin proteins [30]. In addition, membrane vesicles isolated from broccoli can act not only as carriers but also have activity on their own, such as specific bioactivity due to glucosinolates, isothiocyanates, or certain proteins present in the membranes after the isolation procedure [30].

The aim of this study was advance in the knowledge previously determined [30] and analyze the potential clinical applications of broccoli membrane vesicles and SFN, either free or encapsulated, studying the effect on the activity of human M1 and M2 macrophages. Therefore, we established in vitro primary cultures of monocyte-derived macrophages obtained from peripheral blood of healthy donors that were differentiated toward either proinflammatory or M1 or anti-inflammatory or M2 cell profile populations. The cytotoxicity, the production of proinflammatory (TNF-α, IL-6, and IL-1β) and anti-inflammatory (IL-10) cytokines, and the phagocytosis capability on *Candida albicans* yeast were analyzed on the cell cultures in response to the treatment with the *Brassica* bioactive compounds.

## 2. Results

### 2.1. Characterization of Cytokine Profile of M1 and M2 Monocyte-Derived Human Macrophage Cultures

Primary cell cultures of human macrophages differentiated toward a proinflammatory M1 and an anti-inflammatory M2 profiles were established from peripheral blood monocytes from healthy donors [6]. After the differentiation period in presence of 50 ng/mL of human granulocyte–macrophage colony-stimulating factor (hGM-CSF) or human macrophage colony-stimulating factor (hM-CSF) to obtain M1 and M2 profile, respectively [6], the cells remained alive and adhered to the cell wells, showing the typical morphology of macrophages. The analysis of cytokine production revealed that M1-polarized macrophages secreted detectable levels of the proinflammatory cytokines TNF-α and IL-6, whereas IL-1β and the anti-inflammatory cytokine IL-10 were undetectable (Table 1). As it could be expected for M1 macrophages, the stimulation with lipopolysaccharide (LPS) (0.1 µg/mL) for 24 h, induced a potent increase of the proinflammatory cytokine levels, TNF-α (455-fold increase), IL-6 (1325-fold), and IL-1β, inducing also detectable levels of IL-10 (Table 1).

In contrast to M1 macrophages, M2-polarized cells secreted detectable levels of the anti-inflammatory cytokine IL-10, while the levels of proinflammatory cytokines were low (IL-6), or even undetectable (TNF-α and IL1-1β) (Table 1).

### 2.2. Brassica-Derived Compounds Does Not Induce Cytotoxicity in Primary Cultures of Human M1 Macrophages

To exclude a possible cytotoxic effect of the *Brassica* bioactive compounds on primary cultures of human M1 macrophages derived from peripheral blood monocytes, metabolic activity, and, therefore, cell viability were tested using the MTT assay.

The assay was carried out after 24-h exposure to broccoli membrane vesicles and SFN, free or encapsulated in the broccoli membrane vesicles, in primary M1 macrophage cultures under basal and proinflammatory conditions induced by stimulation with LPS (0.1 µg/mL), simulating the microenvironment of chronic inflammatory pathologies (Figure 1).

Cell cultures of human M1 macrophages in basal conditions (Figure 1a) did not present any significant change on their metabolic activity in response to the treatment with *Brassica* compounds in any of their formulations. When the cells were cultured under inflammatory conditions (Figure 1b), only SFN-loaded broccoli membrane vesicles treatment induced a small but significant reduction of about 20% in their metabolic activity. In addition, cell morphology and the adhesion capability of the M1 macrophages did not vary after those treatments.

### 2.3. Sulphoraphane Induced the Opposite Effect to Unloaded Broccoli Membrane Vesicles on Proinflammatory Cytokine Production at Basal State in M1 Human Macrophages

The analysis of cytokine levels released by in vitro differentiated M1 human macrophages at basal state after 24 h exposure to different *Brassica*-derived compounds revealed that free SFN was able to induce a potent reduction of TNF-α (80%) and IL-6 (40%) levels. (Figure 2). Encapsulation of SFN in broccoli membrane vesicles did not modify significantly the reduction induced by free SFN on TNF-α and IL-6 levels. Under these experimental conditions, the unloaded membrane vesicles showed the opposite effect, i.e., increasing both TNF-α (26-fold increase) and IL-6 (10-fold increase) levels respect to the untreated controls (Figure 2a,b).

### 2.4. Under Inflammatory Environment, Sulforaphane Maintains Its Inhibitory Effect on Cytokine Production Whereas Unloaded Membrane Vesicles Lose Their Activating Effect on M1 Macrophages

Under inflammatory conditions induced by stimulation with LPS of the cell cultures, both free and SFN encapsulated in broccoli membrane vesicles, maintained the effect observed in basal conditions on TNF-α, IL-6 (Figure 3a,b), exerting a potent reduction with a total blockade in the cytokine production, not only for all the proinflammatory cytokines assayed, including IL-1β (Figure 3a–c), but also for the anti-inflammatory cytokine, IL-10 (Figure 3d). In these experimental conditions, unloaded broccoli membrane vesicles did not modify the production of none of the proinflammatory cytokine tested (Figure 3a–c), however, inducing a significant reduction of almost 50% respect to the control levels of the anti-inflammatory cytokine IL-10 (Figure 3d).

### 2.5. Sulphoraphane Reduces the Phagocytic and Lytic Capacity of M1 Macrophages on C. albicans

Phagocytosis, one of the main activities of M1 macrophages, depends on the recognition of microbial antigens through the pathogen-associated pattern recognition receptors (PRRs). We analyzed the effect of treatment with free or encapsulated SFN on the in vitro phagocytic and lytic capacity of M1 macrophages. For this, the macrophages were treated with SFN in its different formulations for 24 h, then the medium was removed and replaced by fresh medium to subsequently coculture the macrophages with the yeast *C. albicans*, either alive or heat killed, for 2 h at a 1:5 macrophage to yeast ratio.

Similar results were obtained either under basal or inflammatory conditions. The images obtained from optical microscope inspection of the cocultures showed that human M1 macrophages actively phagocyted *C. albicans* yeast cells (Figure 4). It was observed that unloaded broccoli membrane vesicles did not modify the phagocytic ability (Figure 4b), showing similar rate of phagocytosing macrophages and ingested yeast compared with controls (Figure 4a). The treatment with SFN reduced the phagocytic ability (Figure 4c,d), being this effect more evident in the encapsulated format, where none of the macrophages ingested yeast which remained outside the cells (Figure 4d).

When the experimental procedures were performed with live *C. albicans* yeast from exponential phase cultures, similar results were obtained (Figure 5). *C. albicans* is capable to modify its morphology in response to environmental changes that promote its survival. The yeast-to-hypha transition is enhanced by several external stimuli that mimic the host environment, as temperature of 37 °C or serum exposure, which is a well-known mechanism to evade phagocytosis [31,32]. As shown in Figure 4a, cells from the control group could recognize and endocytose most of yeasts. Some of the yeasts, mainly those that were not ingested by macrophages, developed hyphae. Macrophages previously treated with *Brassica* compounds, in all formulations tested, were less efficient than controls in terms of phagocytosis and lysis of yeasts, eliciting their growth and hyphae formation. The reduction in phagocytosis elicited by SFN, either free or encapsulated (Figure 5c,d, respectively) was stronger than the one mediated by unloaded broccoli membrane vesicles (Figure 5b), in which the highest number of yeasts with the largest hyphal formation were recorded.

The interaction of M1 macrophages with *C. albicans* for 2 h induced the release of proinflammatory cytokines (Table 2). Thus, in basal state, the levels of TNF-α in response to *C. albicans* increased 116-fold, the levels of IL-6 reached 404-fold activation, whereas IL-1β was undetectable. Under inflammatory conditions, the increases of cytokine levels registered in response to *C. albicans* were weaker, reaching to 3.2-fold increase for TNF-α, 1.6-fold for IL-6. The IL-1β production levels in response to *C. albicans* were similar to the control values.

Pretreatment of M1 macrophages with SFN-loaded broccoli membrane vesicles produced a complete blockade of *C. albicans* mediated induction of TNF-α, IL-6, and IL-1β in all experimental procedures tested (Figure 6). Free SFN had also a potent effect that was more evident under the inflammatory conditions elicited by LPS stimulation where the levels of IL-1β and IL-6 were almost undetectable, and the reduction of TNF-α levels were 88% compared with untreated controls. Pretreatment with free SFN at basal state induced a reduction of 91% for IL-6 and 52% for TNF-α respect to the control levels. Pretreatment with unloaded broccoli membrane vesicles had a weaker effect, showing only a significant reduction at basal state for IL-6 levels, and under inflammatory conditions for IL-1β (Figure 6b,e).

### 2.6. Sulphoraphane Modulates the Production of IL-10 in M2 Human Macrophages

The cell viability of human M2 macrophages (Figure 7a) did not vary in response to the presence of *Brassica* compounds treatments in any of their formulations. The IL-10 secretion (Figure 7b) was significantly reduced by SFN, being the effect stronger when it was encapsulated in broccoli membrane vesicles. Nevertheless, the unloaded broccoli membrane vesicles had the opposite effect, inducing a significant increase in the IL-10 production levels.

## 3. Discussion

In this study, we have analyzed the effect of SFN free or encapsulated in broccoli membrane vesicles on the activity of human macrophages primary cultures. Physiologically, tissue resident macrophages display a continuous flux of phenotypes and differentiation or polarization profiles, which allow them to constantly adapt to their environment. The alteration of this homeostatic macrophage balance has been related with several diseases [8,10,33]. Different in vitro methods can be found in the literature to obtain a concrete polarization phenotype of macrophages from blood monocytes. Herein, primary cell cultures of differentiated human macrophages toward a proinflammatory M1 and anti-inflammatory M2 polarization profile were established from peripheral blood monocytes from healthy donors, following the protocol described by Carvalheiro et al. [6]. After differentiation, the cells remained viable, showed the typical morphology of macrophages, and fulfill their correspondent function, producing the characteristic cytokines of their differentiation profile (M1 cells produced IL-6 and TNF-α, while M2 cells secreted IL-10 and IL-6, in basal state) and were able of detect, internalize, and degrade microorganisms (M1 macrophages). This in vitro cell model allows us to analyze the potential anti-inflammatory activity of the compounds studied in basal state and under a low-grade inflammatory scenario induced by stimulation with LPS, which mimics the inflammatory conditions present in chronic inflammatory diseases. This experimental model for drug screening has several advantages compared to others, since our cells do not divide indefinitely as they are not tumor cell lines, so they are more similar to physiological conditions; and on the other hand, the method to obtain blood monocytes is easier, less aggressive, and yields a greater number of cells compared to the isolation of tissue macrophages. Nevertheless, since the phenotype of the macrophages obtained could differ based on the different protocols available in literature, the detailed definition of such protocols and the use of a normalized nomenclature for these cell types, could avoid the controversy generated at this regard [34]. Meanwhile, the use of these models remains a useful experimental tool to obtain macrophages derived from monocytes in pro- or anti-inflammatory profiles.

Our results have shown that treatment with broccoli compounds for 24 h did not induce cytotoxic effect in any of their formulations on the monocyte-derived M1 and M2 macrophage primary cultures. Only treatment with SFN encapsulated in broccoli membrane vesicles seemed to present a slight decrease in their metabolic activity, although it does not represent any inconvenience for its use as a possible therapeutic tool for this cell type.

The intensity in the effect of *Brassica* compounds seemed to differ depending on the inflammatory state of the cells. Treatment with 25 µM SFN for 24 h exerted a strong decrease in the levels of pro-inflammatory cytokines produced by M1 macrophages, both in basal conditions (TNF-α and IL-6) and in a low-grade inflammatory scenario induced by stimulation with LPS (TNF-α, IL-6, and IL-1β), in which the effect was even stronger, inducing a total blockade of the cytokine secretion into the extracellular milieu. Encapsulation of SFN in broccoli membrane vesicles has been reported to improve the stability and half-life of SFN, and thus exhibit a greater effect [23,29]. Nevertheless, in this experimental model, we found no differences between free SFN and SFN in the encapsulated format, probably due to the powerful effect shown by the free compound. These results are consistent with those obtained previously for TNF-α and IL-6 in macrophage-like cell line models with free SFN [35] or SFN encapsulated in broccoli membrane vesicles [30], although the effect recorded was less potent. The effect previously described on IL-1β in cell line models could only be detected when SFN was loaded into broccoli membrane vesicles, achieving a 50% decrease relative to controls cells under inflammatory conditions [30], whereas the results presented in this study showed that both free and encapsulated SFN completely abrogated the effect of IL-1β production of this cytokine in primary cultures of M1 macrophages. These results are also in accordance with previous studies in rats with sciatic endometriosis, where SFN treatment reduced TNF-α, IL-6, and IL-1β. As a result, SFN treatment reduced endometrophic injury due to reduced inflammation [36].

Broccoli membrane vesicles also contain a reduced amount of isothiocyanates such as SFN and other molecules in their own structure with possible bioactivity, the anticancer activity described in a melanoma cell line [30,37]. Therefore, they could represent not only a simple vehicle for drug delivery but also a drug by themselves. Our results showed that treatment with unloaded broccoli membrane vesicles had the opposite effect than free SFN and SFN-loaded broccoli membrane vesicles, inducing a significant increase in TNF-α (26-folds) and IL-6 (10-folds) levels under basal conditions. Many phytochemicals, including SFN, demonstrate a biphasic dose–response relationship and are considered hormetic compounds, i.e., they induce biologically opposite effects at different doses [38]. These hormetic responses are mediated by the activation of nuclear factor erythroid-derived 2 (Nrf2) and antioxidant response elements (AREs). Because of this, the hormetic response is characteristically biphasic, well integrated, concentration-/dose-dependent, and specific with regards to the target cell type and the temporal profile of the response [39,40,41]. Thus, the low concentration of SFN or other bioactive molecules present in unloaded broccoli membrane vesicles could be responsible for the proinflammatory effect, as opposed to the anti-inflammatory activity induced by higher doses of SFN. Again, the effect of *Brassica* compounds seemed to differ according to the inflammatory state of the cells, since under inflammatory conditions, the levels of cytokines in the presence of unloaded broccoli membrane vesicles remained similar to the controls. This could be explained, at least in part, by a possible exhaustion or saturation of the inflammatory response of M1 macrophages under inflammatory conditions, as LPS treatment induced a sharp increase in cytokine production (455-folds increase for TNF-α and 1325-folds for IL-6). On the other hand, the way in which vesicles are incorporated into cells could also play a role in its behavior. In this regard, broccoli membrane vesicles can fuse with the plasma membrane of human keratinocytes [26], delivering the drug into the cell where they are loaded. Nevertheless, phagocytosis must be taken into consideration as a possible pathway for vesicles to enter macrophages [42]. In this case, stimulation with LPS could alter the expression of the membrane receptors involved in the phagocytosis of vesicles, thus modifying the sensitivity to treatment. As described above, IL-10 is an important anti-inflammatory cytokine produced and secreted by LPS-activated macrophages to limit inflammatory responses. Experimental data from a mouse model describe two ways of IL-10 trafficking and release, one of which is related to that of TNF-α and IL-6. Under our experimental conditions, free and encapsulated SFN and unloaded broccoli membrane vesicles inhibited LPS-induced production of the anti-inflammatory cytokine IL-10 in M1 macrophages [43]. Subedi et al. described that 10 µM SFN inhibited the production of pro-inflammatory cytokines (TNF-α, IL-6, and IL-1β); however, in disagreement with our results, they described an increase in the levels anti-inflammatory cytokines (IL-10 and IL-4) in the microglial cell line BV2 activated by LPS [44]. Similar results were obtained by Ali et al. [45], who described that SFN treatment decreased TNF-α, IL-6, IL-23, and IL-1β mRNA expression and increased IL-10 levels in THP-1 macrophage-like cell line. However, consistent with our results, they also found that SFN treatment decreased IL-10 levels when cells were stimulated with LPS and interferon (IFN)-γ for 48 h. Thus, the variability in these results could be due to differences in the SFN dose and drug sensitivity of each cell model.

The phagocytic ability of M1 macrophages on *C. albicans* was reduced after 24 h of treatment with broccoli compounds on in vitro cultures. Pretreatment with the broccoli compounds yielded similar results for M1 macrophages in the basal state or under inflammatory conditions. Untreated M1 macrophages were able to recognize and engulf heat-killed yeasts, finding no differences in cells treated with unloaded broccoli membrane vesicles respect to the control cells. However, the pretreatment with SFN reduced the ingestion of heat-killed *C. albicans*, being more evident for the encapsulated vesicle format. Phagocytosis assays performed with live yeast revealed similar results, although in this case, unloaded broccoli membrane vesicles were also able to reduce phagocytosis compared with untreated control cells. The different set of interactions that can take place between PAMPs expressed in the yeast cell wall, such as β-glucans, with its specific PRR exposed on the macrophage cell surface, such as dectin-1 [46], could contribute to the differences found. Consistent with our results, it has been reported that SFN treatment was able not only to reduce the production of proinflammatory cytokines but also to suppress the antibody-independent phagocytic and chemotactic migratory abilities of human peripheral blood monocytes [47]; and to reduce the phagocytosis and microbicidal activity of human M1 monocyte-derived macrophages and THP-1 derived macrophages on *Streptococcus aureus* and *Escherichia coli* [45]. It has also been reported that SFN enhanced phagocytosis of polystyrene beads by a murine macrophage-like cell line only in the absence or low concentrations (1%) of fetal bovine serum. However, higher serum concentrations depressed phagocytosis and abolished its stimulation by SFN and did not depend on the induction of Nrf2-regulated genes, as it occurred in peritoneal macrophages of nrf2 (-/-) mice [48]. Opposite results from other studies showed that SFN treatment induced the phagocytic activity of amyloid beta oligomers in microglial cells [49]. Other authors described, in an in vitro model, that activation of the Nrf2 pathway induced by SFN restored the defective phagocytic capacity of alveolar macrophages isolated from patients with chronic obstructive pulmonary disease. However, in a clinical trial in which SFN was administered orally to these patients, such improvement was not observed [50]. Our results showed that the recognition of *C. albicans* by M1 macrophages induced a proinflammatory response, increasing TNF-α, IL-6, and IL-1β levels, which was abolished in cells pretreated with free or encapsulated SFN. After withdrawal of broccoli compounds, M1 macrophages pretreated in the basal state produced lower levels of cytokines than those under inflammatory conditions, in the absence of yeast, indicating the persistence of intracellular events triggered by the treatments. Furthermore, the reduction in the interaction with the yeast observed in macrophages pretreated with free or encapsulated SFN involved in part the reduction in yeast-induced cytokine production.

We reported for the first time the effect of broccoli compounds on human M2 macrophages in vitro. Our results showed a complete blockade of IL-10 production after 24 h of exposure with free SFN and SFN-loaded in broccoli membrane vesicles. This effect represents a new opportunity for clinical application for diseases in which the activity of M2 macrophages must be controlled, such as endometriosis where M2 macrophages support the growth of endometrial lesions contributing to the disease progression [8], or cancer where tumor-associate macrophages, similar in phenotype to M2 macrophages, promote tumor growth and metastasis and are associated with poor prognosis of tumors [5], among others.

In summary, our results show that free and encapsulated SFN induced a potent reduction on the activity of human M1 and M2 monocyte-derived macrophages, downregulating proinflammatory and anti-inflammatory cytokines, and phagocytic capability on *C. albicans*. The broccoli membrane vesicles do not represent inert nanocarriers, as they have low amounts of bioactive compounds, being able to induce opposite effects to that of higher doses of SFN under certain experimental conditions, reflecting its hormetic effect. These data reinforce the potential use of broccoli compounds as therapeutic agents not only for inflammatory diseases, but they also open clinical possibilities for applications in other diseases related to immunodeficiency, autoimmunity, or in cancer therapy. Considering the variability of their biological effects in different scenarios, a proper therapeutic strategy should be designed for each pathology, using different combinations, concentrations, and time patterns of SFN treatments, which can be optimized with different nanocarriers and delivery systems.

## 4. Materials and Methods

### 4.1. Healthy Donors

This study included blood samples from 7 healthy volunteers from the Hemodonation Center of the Region of Murcia. The blood samples were initially aimed at using in blood transfusions; however, it was discarded for not reaching or exceeding the optimal blood transfusion bag weight.

### 4.2. Isolation of Human Peripheral Blood Monocytes

Peripheral blood mononuclear cells (PBMC) fractions were isolated from peripheral blood by density centrifugation with Lymphoprep (Axis-Shield PoC As, Oslo, Norway). Briefly, blood samples were diluted 1:1 with phosphate buffered saline (PBS, Gibco Invitrogen from Thermo Fisher Scientific, Kandel, Germany). Subsequently, 35 mL of diluted blood was added onto 15 mL of Lymphoprep, and centrifugated at 1200 rpm for 30 min at room temperature. Afterward, PBMCs portion was collected by aspiration with a Pasteur pipette and washed twice with PBS. The cell concentration was determined with a Neubauer hemocytometer chamber, discarding the dead cells by trypan blue exclusion test.

In order to isolate the monocytes from PBMCs portion, 30 × 10^6^ cells were seeded in 10 cm culture-treated dishes (Thermo Fisher Scientific, Kandel, Germany), adding 10 mL of Monocyte Attachment Medium (Promocell, Heidelberg, Germany) to promote monocyte adhesion. After 1.5 h of incubation at 37 °C and 5% CO_2_, the supernatant was aspirated from the dishes and vigorously washed 3 times with PBS to eliminate nonadherent cells.

### 4.3. Macrophage Polarization and Cell Culture Preparation

The monocytes previously isolated in 10 cm culture dishes were incubated with 10 mL of complete culture medium (CCM), consisting in DMEM High Glucose medium (Biowest, Nuaillé, France) supplemented with 10% (v/v) of fetal bovine serum (Biowest, Nuaillé, France) and 1% (v/v) of penicillin–streptomycin (Capricorn Scientific, Ebsdorfergrund, Germany). For differentiation toward proinflammatory M1-like macrophages, blood monocytes were stimulated with 50 ng/mL of hGM-CSF (Proteintech, Manchester, UK). To obtain anti-inflammatory M2-like macrophages, blood monocytes were stimulated with 50 ng/mL of hM-CSF (Proteintech, Manchester, UK). The cells were incubated for 7 days at 37 °C and 5% CO_2_, refreshing the cell culture at day 4 with 5 mL of CCM and 50 ng/mL of the corresponding cytokine. After 7 days, the macrophages were washed twice with PBS and subsequently were detached adding 10 mL of Macrophages Detachment Medium (Promocell, Heidelberg, Germany). The dishes were incubated for 40 min at 4 °C and 20 min at room temperature, and afterward, 10 mL of DMEM High Glucose were added to finally detach the cells with the help of a micropipette. Cellular suspension was washed twice with DMEM High Glucose, and the cell concentration was determined with a Neubauer hemocytometer chamber, discarding the dead cells by trypan blue exclusion test.

The nomenclature proposed for these macrophage populations, according with the Murray’s et al. review [34], should be M1 (GM-CSF) or M1-like and M2 (M-CSF) or M2-like. In order to simplify the terminology, they are named M1 and M2 throughout the text.

The macrophages obtained were cultured in 96-well plates at 1.25 × 10^5^ cells/well (Thermo Fisher Scientific, Kandel, Germany) in 200 µL of CCM and incubated for 24 h at 37 °C and 5% CO_2_ to facilitate their adhesion.

### 4.4. Preparation of the Compounds to Be Tested in Cell Culture

The SFN (R,S-sulforaphane, Quimigen SL, Madrid, Spain) was encapsulated in membrane vesicles isolated from *Brassica oleracea* L. var. *Italica* as previously described [30]. In order to achieve optimal sterility and a homogeneously sized population of vesicles, both unloaded broccoli membrane vesicles and SFN-loaded broccoli membrane vesicles were filtrated trough a 0.22 µm Ø PVDF filter (Millipore, Darmstadt, Germany). The compounds were diluted in CCM to reach final concentrations of 25 µM for SFN, 0.02 mg/mL for membrane vesicles, and 25 µM-0.02 mg/mL for SFN-loaded broccoli membrane vesicles.

### 4.5. Macrophage Treatment and Stimulation with LPS

All monocyte-derived macrophages were studied in basal condition, whereas M1 macrophages were also stimulated with bacterial lipopolysaccharide (LPS) (*Escherichia coli* 0111.B4; Sigma Chemical Co., St. Louis, MO, USA) to induce a proinflammatory environment.

Both M1 and M2 macrophages were treated with the different compounds (free SFN, unloaded broccoli membrane vesicles, and SFN-loaded broccoli membrane vesicles). After 30 min of incubation at 37 °C and 5% CO_2_, LPS (0.1 µg/mL) was added to the corresponding wells. The cells were incubated for 24 h, and the supernatant was removed and stored at −20 °C for cytokine measurement. The cells were then used for viability or phagocytosis assays.

### 4.6. Cell Viability Assay

Cell viability of treated macrophages was evaluated by MTT assay, based in the reduction of 3-(4,5-dimethylthiazol-2-yl)-2,5-diphenyltetrazolium bromide to formazan by cellular mitochondrial enzymes [51]. Briefly, MTT 0.3 mg/mL (Alfa Aesar, Thermo Fisher, Karlsrhue, Germany) was added to the macrophages and the plates were incubated at 37 °C and 5% CO_2_ for 2 h. Then, lysis solution (isopropanol at 0.01% NP-40 and 0.04 M HCl) was added to solubilize the formazan. Finally, the absorbance at 550 nm was measured by the SPECTROstar Nano (BMG LABTECH, Ortenberg, Germany). The percentage of cell viability was obtained using the following equation:(1)$$\mathrm{Cell}\;\mathrm{viability}\;\left(\%\right)=\frac{{\left({\mathrm{Abs}}_{550\mathrm{nm}}\right)}_{\mathrm{sample}\;}}{{\left({\mathrm{Abs}}_{550\mathrm{nm}}\right)}_{\mathrm{control}\;}}\times 100$$

### 4.7. Phagocytosis Assay

*Candida albicans* cells, SC5314 strain, were grown in Yeast Peptone Dextrose (YPD)-agar medium consisting of 2% peptone, 1% yeast extract, 2% glucose, and 2% agar. They were maintained by periodic subculturing in solid YPD-agar. Exponential cultures grown for 4 h at 37 °C, were washed with PBS, resuspended in CCM, and afterward, cell concentration was determined with a Neubauer hemocytometer chamber. *C. albicans* yeast cells were cocultured with the M1 macrophages pretreated for 24 h with broccoli compounds in 1:5 macrophage:yeast proportion at 37 °C and 5% CO_2_ for 2 h. After incubation time, supernatant was removed and stored at −20 °C until cytokine measurement, whereas macrophages–yeasts were washed once with PBS and photographed using a Leica DMi8 inverted fluorescence microscope coupled with DFC3000G camera (Leica Microsystems, Mannheim, Germany).

### 4.8. Cytokine Quantification

TNF-α, IL-1β, IL-6, and IL-10 cytokines from macrophages cultures were quantified by enzyme-linked immunosorbent assay (ELISA) according to the manufacturer’s instructions (Invitrogen, Thermo Fisher Scientific, Waltham, MA, USA). The assays were performed in strip plates (Immuno Clear Standard Modules, Thermo Scientific, Waltham, MA, USA), and the final absorbance at 470 nm and 550 nm was measured by the SPECTROstar Nano (BMG LABTECH, Ortenberg, Germany). The cytokines concentration was calculated using the corresponding standard curve, and cytokine levels were normalized by referring the results obtained to control values in each experiment, which were valued as 100.

### 4.9. Statistical Analysis

GraphPad Prism version 9.3.0 (GraphPad, Chicago, IL, USA) was used to analyze the data obtained in this work. Data were normalized relative to the untreated control, which was valued as 100%. Statistical differences between normalized data were calculated by one-way ANOVA test followed by a Tukey’s test as a post hoc analysis. Statistically significant differences were obtained when *p* < 0.05. All experiments were performed at least, in duplicate, with three healthy donors and the results are showed as the mean ± SEM.

## Figures and Tables

**Figure 1 ijms-23-11141-f001:**
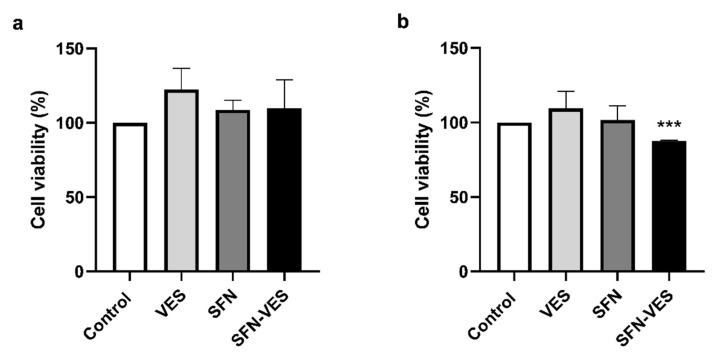
Analysis of cytotoxicity of *Brassica*-derived compounds on primary cultures of human M1 macrophages. Unstimulated (**a**) and LPS (0.1 µg/mL) stimulated (**b**) M1 macrophages were untreated (control) or treated for 24 h with unloaded broccoli membrane vesicles 0.02 mg/mL (VES), free sulforaphane 25 µM (SFN), or SFN-loaded broccoli membrane vesicles at 25 µM (SFN-VES). Viability was assessed by MTT assay in M1 macrophages from 6 healthy donors. Data were normalized relative to the untreated control for each experimental condition (valued as 100%) and represented as mean ± SEM. Significant differences with respect to the untreated control (*) was calculated according to the one-way ANOVA analysis with a Tukey’s test as a post hoc test (*** *p* < 0.001).

**Figure 2 ijms-23-11141-f002:**
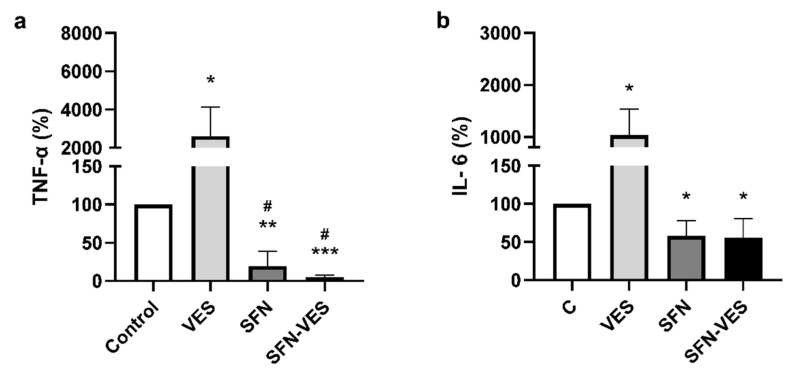
Modulation of human M1 macrophages cytokines production by treatment with *Brassica*-derived compounds. Unstimulated primary cultures of human M1 macrophages were untreated (control) or treated for 24 h with, unloaded broccoli membrane vesicles 0.02 mg/mL (VES), free sulforaphane 25 µM (SFN), or SFN-loaded broccoli membrane vesicles at 25 µM (SFN-VES). TNF-α (**a**) and IL-6 (**b**) cytokine production was assessed from M1 macrophages obtained from 5 healthy donors. Data were normalized relative to the untreated control for each experimental condition (valued as 100%) and represented as mean ± SEM. Significant differences with respect to the untreated control (*) or VES treatment (#) were calculated according to the one-way ANOVA analysis with a Tukey’s test as a post hoc test (* *p* < 0.05, ** *p* < 0.01, *** *p* < 0.001, # *p* < 0.05).

**Figure 3 ijms-23-11141-f003:**
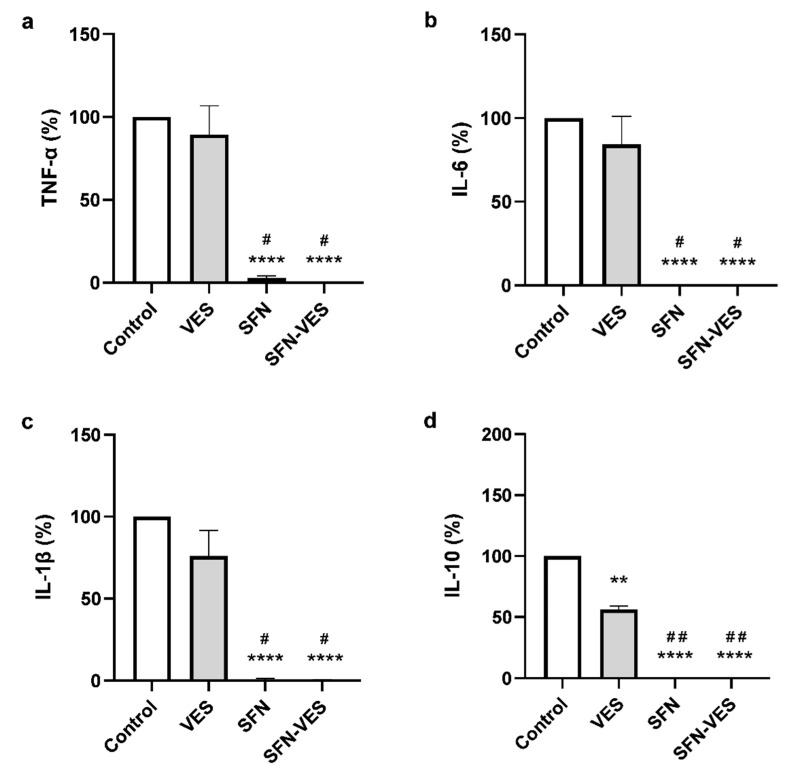
Modulation of cytokine production by treatment with *Brassica* derived compounds in human M1 macrophages. LPS stimulated primary cultures of human M1 macrophages were untreated (control) or treated for 24 h with, unloaded broccoli membrane vesicles 0.02 mg/mL (VES), free sulforaphane 25 µM (SFN), or SFN-loaded broccoli membrane vesicles at 25 µM (SFN-VES). TNF-α (**a**), IL-6 (**b**), IL-1β (**c**) and IL-10 (**d**) cytokine production was assessed from M1 macrophages obtained from 5 healthy donors. Data were normalized relative to the untreated control for each experimental condition (valued as 100%) and represented as mean ± SEM. Significant differences with respect to the untreated control (*) or VES treatment (#) were calculated according to the one-way ANOVA analysis with a Tukey’s test as a post hoc test (** *p* < 0.01, **** *p* < 0.0001; # *p* < 0.05, ## *p* < 0.01).

**Figure 4 ijms-23-11141-f004:**
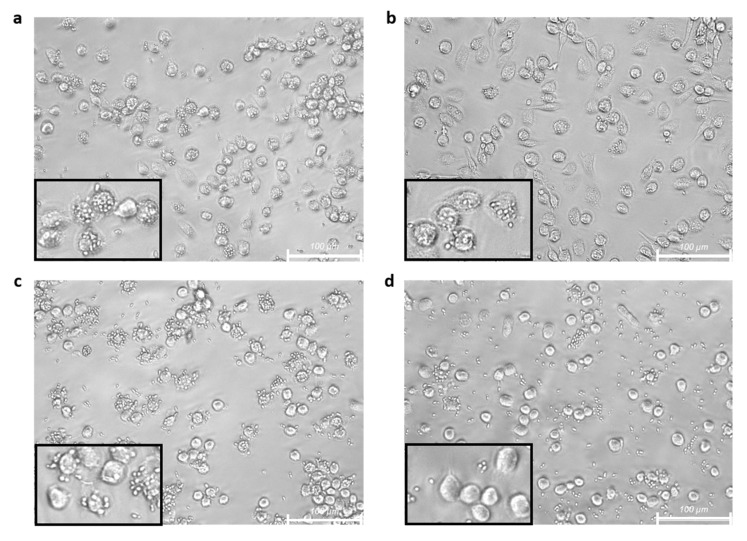
Phagocytosis of heat killed *C. albicans* mediated by M1 macrophages after treatment with *Brassica*-derived compounds. LPS stimulated (0.1 µg/mL) human M1 macrophages from 3 healthy donors were untreated (**a**) or treated for 24 h with unloaded broccoli membrane vesicles 0.02 mg/mL (**b**), free sulforaphane 25 µM (**c**), or SFN-loaded broccoli membrane vesicles at 25 µM (**d**). After withdrawal of medium, the cells were cocultured for 2 h with heat killed *C. albicans* at 1:5 ratio, and then washed with PBS. Representative images from optical microscope with 20× magnification are shown. Bar scale is 100 µm.

**Figure 5 ijms-23-11141-f005:**
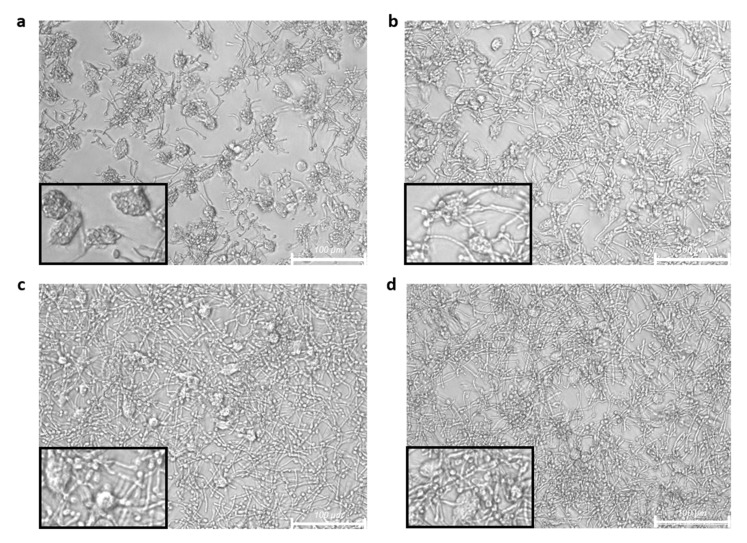
Phagocytosis of *C. albicans* mediated by M1 macrophages after treatment with different formulations of SFN. LPS stimulated (0.1 µ/mL) human M1 macrophages from 4 healthy donors were untreated (**a**) or treated for 24 h with unloaded broccoli membrane vesicles 0.02 mg/mL (**b**), free sulforaphane (SFN) 25 µM (**c**), or SFN-loaded broccoli membrane vesicles at 25 µM (**d**). After withdrawal of medium, the cells were cocultured for 2 h with *C. albicans* from exponential phase cultures, at 1:5 ratio and then washed with PBS. Representative images from optical microscope with 20× magnification are shown. Bar scale is 100 µm.

**Figure 6 ijms-23-11141-f006:**
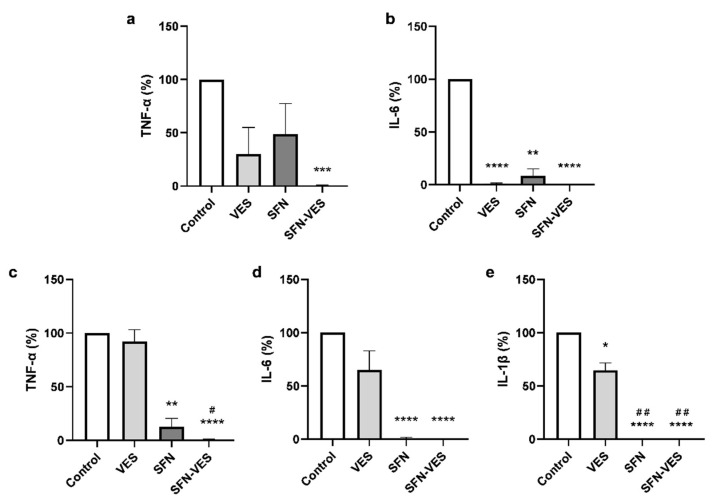
Cytokine production by M1 macrophages treated with SFN in its different formulations cultured in the presence of *C. albicans*. Unstimulated (**a,b**) or LPS stimulated (**c–e**) primary cultures of human M1 macrophages from 3 or 5 healthy donors, respectively, were untreated (control) or treated for 24 h with unloaded broccoli membrane vesicles 0.02 mg/mL (VES), free sulforaphane 25 µM (SFN), or SFN-loaded broccoli membrane vesicles at 25 µM (SFN-VES). After withdrawal of medium, the cells were cocultured for 2 h with *C. albicans* from exponential phase cultures, at 1:5 ratio (macrophage:yeast). Cytokine production was assessed in culture supernatants. Data were normalized relative to untreated control for each experimental condition (valued as 100%) and represented as mean ± SEM. Significant differences with respect to the untreated control (*) or VES treatment (#) were calculated according to the one-way ANOVA analysis with a Tukey’s test as a post hoc test (* *p* < 0.05, ** *p* < 0.01, *** *p* < 0.001, **** *p* < 0.0001; # *p* < 0.05, ## *p* < 0.01).

**Figure 7 ijms-23-11141-f007:**
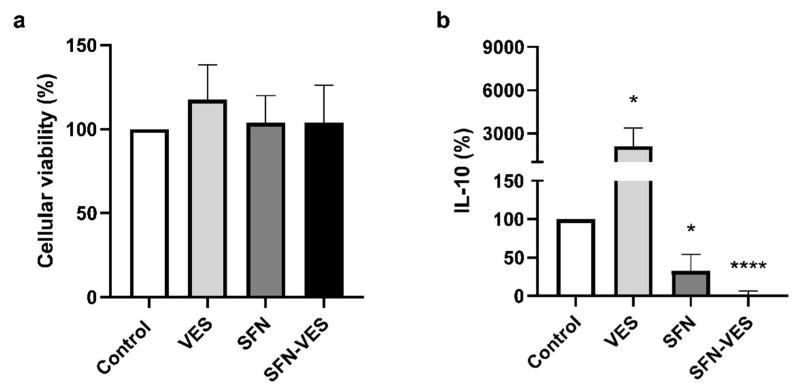
Analysis of viability and secretion of IL-10 in M2 macrophages treated with *Brassica* compounds. Primary cultures of human M2 macrophages were untreated (Control) or treated for 24 h with unloaded broccoli membrane vesicles 0.02 mg/mL (VES), free sulforaphane 25 µM (SFN), or SFN-loaded broccoli membrane vesicles at 25 µM (SFN-VES). Cell viability (**a**) and IL-10 production (**b**) was assessed in M2 macrophages from 4 healthy donors. Data were normalized relative to the untreated control (valued as 100%) and represented as mean ± SEM. Significant differences with respect to the untreated control (*) was calculated according to the one-way ANOVA analysis with a Tukey’s test as a post hoc test (* *p* < 0.05, **** *p* < 0.0001).

**Table 1 ijms-23-11141-t001:** Cytokine production induced in human monocyte-derived M1 and M2 macrophages.

Profile	Treatment	Cytokine (pg/10^6^ cells)			
		TNFα	IL-6	IL-1β	IL-10
**M1**	Control	33.4 ± 10.6	30.7 ± 11.8	n.d.	n.d.
	LPS	15211.0 ± 5318.0	40695.0 ± 11740.0	56.5 ± 32.1	134.5 ± 10.4
**M2**	Control	n.d.	34.8 ± 18.0	n.d.	11.6 ± 2.1

Data represent mean ± SEM of cytokine levels produced in vitro by M1 and M2 monocyte-derived macrophages (pg/10^6^ cells) from 6 healthy donors, cultured for 24 h; n.d. = not detectable.

**Table 2 ijms-23-11141-t002:** Cytokine production induced by *C. albicans* in human monocyte-derived M1 macrophages.

State	Treatment	Cytokine (pg/10^6^ cells)		
		TNF-α	IL-6	IL-1β
Basal	Control	4.8 ± 1.1	1.1 ± 0.8	n.d.
	Control + yeast	563.3 ± 232.1	404.2 ± 167.7	-
Inflammatory (LPS-induced)	Control	229.0 ± 96.3	2075.7 ± 139.5	86.3 ± 46.7
	Control + yeast	729.9 ± 292.6	3349.7 ± 723.2	115.1 ± 59.8

Data represent mean ± SEM of cytokine levels produced in vitro by M1 monocyte-derived macrophages (pg/10^6^ cells) from 5 healthy donors cultured at ratio 1:5 (macrophage:yeast) with *C. albicans* for 2 h; n.d. = not detectable.

## Data Availability

Not applicable.

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
