# Peer review of "Potential of Sulforaphane and Broccoli Membrane Vesicles as Regulators of M1/M2 Human Macrophage Activity"

_ijms, 2022, doi:10.3390/ijms231911141_

Round 1
Reviewer 1 Report
This manuscript indicated M1/M2 macrophage activity regulation through vesicle containing SFN from Broccoli. This result are very interesting. But some corrections may be required. In table 1, level of cytokines was analyzed. In addition, it is better to examine expression levels of other markers of M1/M2 macrophages, such as IL-23, CXCL8, TLR2/4, Ccl2,CD163, CCL5, IL-17, IFN-g, COX2, iNOS, STAT1, Dectin-1, TGF-b, IRF5, SOCS3, etc. In figure 4 and 5, phagpcytosis by M1 macrophage is shown. Further, it is better to analyze detection of phagpcytosis by labelling of phagocytosis particle. In fig 6, it is beteer to analyze cytokine levels by western blot. In discussion, it is better to add limitation(s) and advantage(s) as for analysis of this study. In addition, it is better to examine migration of macrophages with SFN.
Author Response
Answers to the questions raised by the reviewers:
Reviewer: 1
We thank the reviewer very much for his/her valuable comments and suggestions that have improved the understanding of the revised version of our manuscript.
Specific comments for the authors:
This manuscript indicated M1/M2 macrophage activity regulation through vesicle containing SFN from Broccoli. These results are very interesting. But some corrections may be required.
- In table 1, level of cytokines was analyzed. In addition, it is better to examine expression levels of other markers of M1/M2 macrophages, such as IL-23, CXCL8, TLR2/4, Ccl2, CD163, CCL5, IL-17, IFN-g, COX2, iNOS, STAT1, Dectin-1, TGF-b, IRF5, SOCS3, etc.
As the reviewer rightly pointed out, there are many markers to define M1 and M2 macrophages, including cell membrane receptors, intracellular signalling molecules, and soluble mediators. As we were interested in deepening the knowledge on the effect of SFN on the regulation of inflammatory response to assess its proposed potential clinical use in chronic inflammatory diseases, we mainly focused on the effect on cytokine secretion levels. To do this, we studied the main pro-inflammatory (IL-1β, IL-6 and TNF-α) and anti-inflammatory (IL-10) cytokines, describing the basal secretion levels in our study sample, and the effect of Brassica compounds on those cells, comparing the results obtained with other experimental models. In future studies, it could be interesting to address the effect of SFN and broccoli membrane vesicles on other macrophage phenotypic markers, such us some of those suggested by the reviewer.
- In figure 4 and 5, phagocytosis by M1 macrophage is shown. Further, it is better to analyze detection of phagocytosis by labelling of phagocytosis particle.
We agree with the reviewer, as labelling of particles allows discerning between ingestion and adhesion to the cell membrane surface of phagocytes. This is especially necessary when small particles are assayed. However, Candida albicans yeast cells are big enough to be clearly identified under a light microscopy, and although its labelling could help to differentiate between yeast endocytosed or only adhered to the cell surface, the labelling process could also alter the interaction with macrophages, particularly when the yeasts are alive. Thus, the effect of Brassica compounds on the interaction (recognition and endocytosis) between macrophages and yeasts was clearly evident under these experimental conditions.
- In fig 6, it is better to analyze cytokine levels by western blot.
As explained in point 1, we were interested mainly on cytokines released by macrophages which can be efficiently measured by ELISA. Western blot analysis would report some additional information which would be interesting in further studies to deep inside the transcriptional and post-transcriptional regulation mechanisms elicited by broccoli compounds.
- In discussion, it is better to add limitation(s) and advantage(s) as for analysis of this study.
As suggested by the reviewer, we have now included such information in the discussion section in the revised version of the manuscript (lanes 309-319).
- In addition, it is better to examine migration of macrophages with SFN.
The chemokine secretion levels and expression of adhesion molecules, which control migration process, are indeed other interesting candidates to evaluate the effect of Brassica compounds in future works.

Reviewer 2 Report
1. The main limitation of the study is the production of M1 and M2 types of macrophages. Although there are no strict protocols for the production of activated macrophages, authors should read Murray's literature review and the issues discussed there (https://pubmed.ncbi.nlm.nih.gov/25035950/). In this regard, it should be noted that maintaining macrophage identity without MCSF is impossible, while the authors obtained the M1 cell subtype using GM-CSF. Indeed, earlier data have been presented that predifferentiation with GMCSF gives a more stable M1 phenotype (https://www.sciencedirect.com/science/article/abs/pii/S0171298514000862
https://onlinelibrary.wiley.com/doi/full/10.1111/sji.12162), but the pro-inflammatory phenotype itself is obtained when using LPS in combination with IFN-γ. Murray's review generally indicates that the use of GMCSF should be abandoned, as the result is too specific a phenotype. Thus, the authors obtained some predifferentiated macrophage forms, which they call M1 and M2, which is incorrect, and then another type was affected by LPS. In this regard, the following question, why did not study the effect of LPS on macrophages obtained under the influence of MCSF? Given the concept of the phenotypic continuum of macrophages, this also needs to be done.
2. Another issue is statistical processing. In the Materials and Methods section, the authors should indicate more clearly and in detail that the control cultures were taken as 100%. In this regard, it is not clear why the authors compared these indicators using the student's test. What kind of normal distribution of features can we talk about when not natural numbers are compared, but the times in which the indicators changed relative to the control? In addition, in fact, we are dealing with multiple comparisons, when several groups are compared against one control. In this case, it is necessary to use analysis of variance, and since the distribution here is clearly different from normal, it is necessary to use rank analysis of variance with subsequent post hoc comparison using suitable criteria.
Author Response
Answers to the questions raised by the reviewers:
Reviewer: 2
We thank the reviewer very much for his/her valuable comments and suggestions that have improved the understanding of the revised version of our manuscript.
Comments to the Author
- The main limitation of the study is the production of M1 and M2 types of macrophages. Although there are no strict protocols for the production of activated macrophages, authors should read Murray's literature review and the issues discussed there (https://pubmed.ncbi.nlm.nih.gov/25035950/). In this regard, it should be noted that maintaining macrophage identity without MCSF is impossible, while the authors obtained the M1 cell subtype using GM-CSF. Indeed, earlier data have been presented that predifferentiation with GMCSF gives a more stable M1 phenotype (https://www.sciencedirect.com/science/article/abs/pii/S0171298514000862 / https://onlinelibrary.wiley.com/doi/full/10.1111/sji.12162), but the pro-inflammatory phenotype itself is obtained when using LPS in combination with IFN-γ. Murray's review generally indicates that the use of GMCSF should be abandoned, as the result is too specific a phenotype. Thus, the authors obtained some pre-differentiated macrophage forms, which they call M1 and M2, which is incorrect, and then another type was affected by LPS. In this regard, the following question, why did not study the effect of LPS on macrophages obtained under the influence of MCSF? Given the concept of the phenotypic continuum of macrophages, this also needs to be done.
As we were interested in deepening the knowledge about the effect of SFN on the regulation of inflammatory response to evaluate its proposed potential clinical use in chronic inflammatory diseases, we mainly focused on cytokine secretion levels. Previous studies have been mainly performed searching for the capability of the assayed compounds to reduce the inflammatory response in experimental models with pro-inflammatory scenarios. In this work we also pretended to explore the effect in the opposite situation.
To further explore the effect of Brassica compounds we were successfully able to differentiate monocytes from each donor towards two opposite phenotype profiles, proinflammatory or M1 and anti-inflammatory or M2 by incubation with GM-CSF or M-CSF, respectively [6]. This experimental procedure has been previously described and validated by other authors by the characterization of the expression of several M1/M2 markers.
As pointed out by the reviewer, according with what we mentioned in the discussion section, there are many protocols to produce M1 and M2 macrophages in vitro. There is some controversy in the literature about the correct nomenclature for these cell types. In this sense, description of the correct nomenclature suggested by Murray´s review for our monocyte derived macrophage primary cultures, M1 (GM-CSF) or M1-like, and M2 (M-CSF) or M2-like macrophages, is included in material and methods (lines 487-489) and discussion (314-319) sections of the revised version of the article. Murray´s review is the new reference 34, included in references section (line 667-669) in the revised version of the manuscript.
Addition of LPS to M1 (GM-CSF) macrophages stimulated their pro-inflammatory potential, emulating the microenvironment of chronic inflammatory pathologies. This is especially useful for exploring the anti-inflammatory potential of the assayed compounds. Nevertheless, addition of LPS to M2 (M-CSF) macrophages, would repolarize the cells towards a M1-like as previously described (ref. 34, figure 1b). The scope of our work was to analyze the effect of Brassica compounds in opposite pro- and anti-inflammatory conditions, for which we use M1-like and M2-like macrophages.
Even so, after performing the experiments suggested by reviewer, we could confirm the modulation of M2-like macrophages towards a M1 profile after exposition to LPS as detected by the increasing of pro-inflammatory cytokine levels (table 1s) [34]. The effect of broccoli membrane was similar in all macrophage polarization phenotypes assayed, inducing an almost complete blockade in cytokine secretion (Figure 1s). We have not included these results in the revised version of manuscript as supplementary figure, as they would hinder the understanding of our work, distracting from the main objective, which is the study of the effect of Brassica compounds on human macrophages in opposite pro- and anti-inflammatory conditions. In any case, we are open to the reviewer´s suggestions at this regard.
Table 1s. Cytokine production induced in human monocyte-derived M2 macrophages stimulated with LPS.
Profile |
Treatment |
Cytokine (pg/106 cells) |
|||
|
|
TNFα |
IL-6 |
IL-1β |
IL-10 |
M2 |
LPS |
2996.0 ± 1037.7 |
5910.9 ± 1319.4 |
n.d. |
4085.8 ± 698.4 |
Data represent mean ± SEM of cytokine levels produced in vitro by M2 monocyte-derived macrophages (pg/106 cells) from 3 healthy donors, cultured for 24 hours; n.d. = not detectable
Figure 1s. Modulation of cytokine production by treatment with Brassica derived compounds in human M2 macrophages stimulated with LPS. LPS stimulated primary cultures of human M2 macrophages were untreated (control) or treated for 24 hours with, unloaded broccoli membrane vesicles 0.02 mg/mL (VES), free sulforaphane 25 µM (SFN) or SFN‐loaded broccoli membrane vesicles at 25 µM (SFN-VES). Cytokine production was assessed from M2 macrophages obtained from 3 healthy donors. Data were normalized relative to the untreated control for each experimental condition (valued as 100%) and represented as mean ± SEM. Significant differences with respect to the untreated control (*) o VES treatment (#) were calculated according to the one-way ANOVA analysis with a Tukey’s test as a post hoc test (**** p < 0.0001; # p < 0.05)
- Another issue is statistical processing. In the Materials and Methods section, the authors should indicate more clearly and in detail that the control cultures were taken as 100%.
We thank the reviewer for the suggestion. It has been amended in the revised version of the manuscript (line 551- 552).
- In this regard, it is not clear why the authors compared these indicators using the student's test. What kind of normal distribution of features can we talk about when not natural numbers are compared, but the times in which the indicators changed relative to the control? In addition, in fact, we are dealing with multiple comparisons, when several groups are compared against one control. In this case, it is necessary to use analysis of variance, and since the distribution here is clearly different from normal, it is necessary to use rank analysis of variance with subsequent post hoc comparison using suitable criteria.
We thank the reviewer for the suggestion. The results have been re-analyzed with the one-way ANOVA test followed by a Tukey’s test as a post hoc analysis. It has been amended in figure legends and material and methods section in the revised version of the manuscript (lines 552- 553).

Round 2
Reviewer 1 Report
This manuscript eas corrected according to some reviewer's comments.
Reviewer 2 Report
All questions received exhaustive answers, the corrections made are satisfactory.